# How does a recent gender norms scale perform? Exploratory factor analyses among adolescents in Ethiopia and Bangladesh

Anita Alaze[1]*, John Grosser[2], Oliver Razum[1], Céline Miani[1]

1 Department of Epidemiology and International Public Health, School of Public Health, Bielefeld University, Bielefeld, Germany, 2 Department of Health Economics and Health Care Management, School of Public Health, Bielefeld University, Bielefeld, Germany

* anita.alaze@uni-bielefeld.de

## Abstract

Inequitable gender norms shape adolescents' perceptions and behaviours, increasing the risk for adverse health outcomes as adults. However, there is a lack of reliable scales to measure these norms. The Gender and Adolescence: Global Evidence (GAGE) project proposes a scale for adolescents aged 10–19 years considered vulnerable, (i) distinguishing between individual-level gender attitudes and community-level gender norms (2 factors), and (ii) categorising items into five domains (e.g., education; 5 factors). As part of validating this scale, we analyse the two- and five-factor structure using GAGE datasets from Ethiopia and Bangladesh. We performed Explorative Factor Analyses (EFA) using Principal Axis Factoring and oblique rotation. We tested sampling adequacy using Bartlett's test of sphericity and the Kaiser-Meyer Olkin measure. In the EFA, we tested the two-factor structure and refined the initial five-factor structure by removing variables that failed to load onto a factor or exhibited cross-loadings. Next, we removed variables with low communalities (<0.2) and low factor loadings (<0.3). The EFAs comprised 6,183 observations from the Ethiopia and 2,245 observations from the Bangladesh dataset. The initial five-factor solution seemed more appropriate than the two-factor (individual-community) distinction in both datasets, and only the refined five-factor structure provided a solution in which the items corresponded to the five domains. However, this only applies to 17 and 15 of the original 30 items in Ethiopia and Bangladesh, respectively, and two of the factors only include two variables each. The five-domain structure proved more suitable for the Ethiopia and Bangladesh contexts than the individual-community distinction, but only for a reduced set of items. We thus propose an adaptation for the GAGE gender norms scale in Ethiopia and Bangladesh. Conceptual challenges, such as questionable domain assignments, highlight the need to further refine the scale and to confirm the results by Confirmatory Factor Analysis.

**Data availability statement:** The data on which the results of our study are based are the publicly available Gender and Adolescence: Global Evidence (GAGE) data, archived by the data provider 'UK data service' (see https://ukdataservice.ac.uk/). The datasets include the baseline data for Ethiopia and Bangladesh from 2017 and 2018: GAGE Ethiopia baseline data (core respondent); GAGE Bangladesh Dhaka baseline data (core respondent); and GAGE Bangladesh Chitsyl 2018 cross-sectional (core respondent). These datasets are publicly available and can be accessed free of charge after registration with the UK Data Service and the set-up of a research project. We confirm that we did not receive any special privileges in accessing the data that are not available to other researchers.

**Funding:** This paper is part of a PhD and was thereby funded within the scope of the Junior Research Group GendEpi (Gender Epidemiology) within the Department of Epidemiology & International Public Health, School of Public Health, Bielefeld University. The second funding phase of the Junior Research Group was funded by Bielefeld University [grant number not applicable, to CM]. The funders had no role in study design, data collection and analysis, decision to publish, or preparation of the manuscript.

**Competing interests:** The authors have declared that no competing interests exist.

# 1 Introduction

Gender inequity hinders the realisation of human rights and health for all. While Sustainable Development Goal (SDG) 5 drives global efforts to reduce gender inequities, persisting gender inequity has been identified as one of the main obstacles hindering further progress on the other SDGs [1]. Among the biggest barriers to gender inequity are societal gender norms [1]. Gender norms are defined as the "spoken and unspoken rules of societies about the acceptable behaviours of girls and boys […] how they should act, look, and even think or feel" [2].

Gender norms have gained recognition as a public health priority, which increased scientific efforts to adequately measure gender norms [3]. While qualitative methods are typically used to i) gain deeper insights about attitudes toward gender norms, and ii) to inform scale development [4], quantitative gender norms scales serve to measure gender norms as an outcome and as an important factor to behaviour change and health outcomes [3]. They are also used to identify and advance interventions that effectively tackle inequitable gender norms [5]. However, in the translation of qualitative findings to quantitative scales, theoretical constructs (e.g., social norms theory) have so far not been adequately incorporated [4,6].

Moreover, a key research gap around the measurement of gender norms concerns their conceptualisation and validity [7,8]. Various quantitative surveys, such as the World Value Survey and the International Social Survey Programme [8], contain unintentional gender bias in question phrasing and translation, mostly rely on proxies for gender norms, such as social aggregates for individual behaviours and attitudes, and do not contain missing information on gender identity and individuals' gendered experiences (e.g., socially constructed gender roles, gender practices and relationship dynamics) [9]. In addition, many different conceptualisations of gender norms are used in the gender norms scales. This can be seen from the different terminologies used in gender norms scales such as the Gender Attitudes Scale (GAS) by Lundgren et al. [10], the International Men and Gender Equality Survey (IMAGES) by Levtov et al., [11], and the G-NORM scale by Sedlander et al. [6]. The conceptualisation is particularly inconsistent when it comes to distinguishing gender norms from related concepts such as gender roles or gender stereotypes. The former refers to "beliefs about appropriate roles for males and females regarding the division of paid labor, homework, and childcare" (Davis & Greenstein, 2009) and the latter "are widely held, generalised assumptions regarding common traits (including strengths and weaknesses), based on group categorisation" [12,13].

Adolescence has been recognised as a window of opportunity to change inequitable gender norms. It is a critical and intensified period of life [4] in which adolescents' perspectives towards gender norms are shaped. Gender norms also have a notable effect on morbidity and mortality during adolescence. For example, body dissatisfaction as part of gender norm concepts can be associated with gender-related motives of self-harm [14–16], such as exposure to bullying, victimisation, and sexual/physical abuse [14]. Self-harm, in turn, is associated with an increased risk for suicide [17]. Gender norms influence the health trajectories of adolescents throughout a lifetime [18]. Adolescents who adopt gender inequitable attitudes and roles are at greater

risk of experiencing adverse gender-related health outcomes as adults [19]. Examples of this include body dissatisfaction, which can lead to eating or mental disorders [16]. Beliefs in male superiority/entitlement or the justification of violence in the partnership can lead to the perpetuation of intimate partner violence and the reinforcement of power imbalances in society [19]. This makes adolescence an influential period in which health-related outcomes are predetermined and individual attitudes towards gender norms are formed.

Although some gender norms scales have been specifically developed for or adapted to adolescents, such as the Gender Equitable Men (GEM) Scale by Vu et al. [20], Attitudes Towards Women Scale (for Adolescents) by Galambos et al. [21], and the Adolescent Femininity Ideology Scale by Tolman and Porche [22], the age group of adolescents has so far been greatly overlooked in research [18].

A conceptual challenge with those scales arises through the fact that particularly younger adolescents (10–14 years) find it difficult to answer the provided questions. This is because many of them may have not yet engaged in sexual relations or have difficulties imagining abstract situations [4]. For example, the GEM scale was originally developed for individuals aged 15–24 years and covers the four domains of violence, reproductive health and disease prevention, sexuality, and domestic chores and daily life [4,23]. A study in Uganda showed that an adaptation which includes only 16 of the original 24 items across the four domains is needed when validating the GEM scale for younger adolescents [4,20]. Although the Gender Norm Instrument (GNI) of the Global Early Adolescent Study [24], and the Male Role Norms Inventory (MNI) – Adolescent, revised by Levant et al. [25] are gender norms scales developed specifically for early adolescence, further research is needed. In particular, because interventions to tackle inequitable gender norms may be more effective at a younger age [26].

Another concern is that most of those scales have been developed in the Global North. This highlights a research gap for adolescents in Lower and Middle-Income (LMI) countries with different cultural and socioeconomic contexts. Moreover, this not only influences who and what is measured, but also risks perpetuating neocolonial biases and assumptions [9,27]. Additionally, the validity of gender norms scales among marginalized or minoritized adolescents is even less understood [7]. This is because socio-structural disadvantages, such as social class, race, ethnicity, migrant or refugee status, (dis) ability, religion, and area of residence, are areas affected by gender norms [28]. Thus, evidence on the exact pathways of gender norms to health is scarce and the understanding of how adolescents perceive gender norms in cross-cultural settings is limited [4].

The Gender and Adolescence: Global Evidence (GAGE) project as the largest longitudinal study on adolescents aims to address these research gaps in adolescents considered vulnerable in the Global South (e.g., adolescents with disabilities, out of school youth, refugees, married under age 18 etc.) and in the conceptualisation of gender norms. The GAGE project has developed a gender norms scale which includes the aforementioned social norms theory and draws on items from multiple scales: the GAS, the GEM, the IMAGES, and the GNI scale. Some items in the GAGE gender norms scale were also newly introduced. They stem from previously conducted qualitative interviews in the GAGE project [29]. Further details of the individual items and the scales from which they were drawn can be found in Table A.5 in the publication by Baird et al. [29].

As part of the GAGE project, Baird et al. (2019) formed two separate gender norm scores from the GAGE gender norms scale, one for individual-level gender attitudes and the other for perceived community-level gender norms. In their publication, they used these scores to predict physical and mental health in Bangladesh and Ethiopia [29]. Individual-level gender attitudes measure what the adolescent thinks about the gendered attitude while perceived community-level gender norms measure descriptive norms as to "what the [adolescent] believes others do" and injunctive norms as to "what the [adolescent] believes others think that s/he should do" [29,30]. Importantly, the gender norms items in the GAGE gender norms scale are further divided into five domains: [1] education; [2] time use; [3] financial inclusion and economic empowerment; [4] relationships and marriage; and [5] sexual and reproductive health domain [29].

Our analysis draws on the paper of Baird et al. [29] in which the GAGE gender norms scale was introduced. This scale has not yet been adequately statistically validated and makes implicit claims about the factor structure of gender norms that have not been empirically verified. Our aim in this study is therefore to explore the factor structure of the GAGE gender norms scale using Exploratory Factor Analysis (EFA). More specifically, we investigate possible factor structures with two or five factors, which correspond to (i) the distinction between individual-level gender attitudes and perceived community-level gender norms, and (ii) the categorisation of the gender norms items into the five domains present in the GAGE data, respectively. Thus, we compute two- and five-factor structures in our EFA, and compare the results between two countries.

## 2 Materials and methods

### 2.1 Ethical approval

Our study did not require ethical approval, as it is based on the analysis of publicly available secondary panel data that does not contain personal identifiers.

### 2.2 Data

Our statistical analysis draws on quantitative data from the GAGE project, a mixed-methods longitudinal research and evaluation programme. GAGE follows 18,000 adolescents aged 10–19 years as well as other community members over 10 years from 2015 to 2024 [31]. The project aims to discover 'what works' [32] for adolescents considered the most vulnerable. The quantitative survey includes adolescents with disabilities, adolescents who were married before the age of 18, refugees, adolescent mothers and out-of-school youth in the six LMI countries included in the project to address the paucity of data on adolescents considered vulnerable [32]. Besides Rwanda, Nepal, Jordan and Lebanon, the survey also includes Ethiopia and Bangladesh [29,32].

Following Baird et al.'s analysis [29], our study draws on the baseline findings collected by the GAGE project in Ethiopia and Bangladesh in 2017 and 2018. Data collection took place in six locations in Ethiopia and in three locations in Bangladesh. While in Ethiopia and one Bangladesh location, a household census was used to identify potential participants, a school-based census was employed in the other two Bangladesh locations [29]. Further details on data collection procedures are described in Baird et al. [29].

The data on which the results of our study are based are third-party data that are publicly archived by the data provider UK data service. They are available free of charge following registration with the data provider.

### 2.3 Measures

The GAGE gender norms scale in the Ethiopia and Bangladesh datasets used by Baird et al. [29] consists of individual-level gender attitudes and community-level gender norms items, both of which are embedded in five content-related domains (education; time use; financial inclusion and economic empowerment; relationships and marriage; sexual and reproductive health). An example for the *education* domain is: "Girls should be sent to school only if they are not needed to help at home". "Girls and boys should share household tasks equally" is an example for the time use domain and "women who participate in politics or leadership positions cannot also be good wives or mothers" is an example for the *financial inclusion and economic empowerment* domain. The *relationships and marriage* domain is captured, for example, by "adolescent girls should marry before the age of 18 years (legal age)" and the *sexual and reproductive health* domain is captured by questions such as "families should control their daughters' behaviors more than their sons". S1 Table in the supplementary material provides an overview of the GAGE gender noms items and their meaning.

The three items in the *sexual and reproductive health* domain and the one item in the *marriage and relationships* domain in italics could not be included as they were only answered by older adolescents. The gender norms items are three-level variables with '1' indicating agreement, '2' indicating partial agreement and '3' indicating disagreement with the

statement. Following Baird et al., we recoded all items with a reverse question (cr_edu_boysfeelings, cr_fin_girlchance, cr_fin_impwsav, cr_fin_nework, cr_fin_eework, cr_tu_statements1, cr_mar_girlfriend, cr_mar_boyfriend, cr_mar_waitedu, cr_srh_proudbod) to ensure that agreement suggests a gender-unequal response.

## 2.4 Exploratory Factor Analysis

We chose EFA to investigate the underlying factor structures of the GAGE gender norms scale for Ethiopia and Bangladesh. We used Principal Axis Factoring and oblimin rotation (an oblique factor rotation method). Oblique rotation allows non-zero correlation between factors [33], which fits our likely highly correlated gender norms items. We used Bartlett's test of sphericity (test for homoscedasticity based on the Chi-squared distribution) and the Kaiser-Meyer-Olkin (KMO) test (measure of sampling adequacy) to assess the suitability of the datasets for EFA. Datasets are suitable, when the null hypothesis for the first test is rejected and the values for the second test are above 0.5 [34]. We conducted all statistical analyses using the software R (version 4.4.1, R Core Team, 2024).

In their analysis, Baird et al. form two separate gender norm scores (one for individual-level gender attitudes, the other for perceived community-level gender norms) and use these scores as predictors in regression models for physical and mental health [29]. Although they do not say so explicitly, this approach seems to imply a supposed two-factor (individual- vs. community-level) structure in the gender norm data.

To explore the validity of this supposed two-factor structure, we first calculated a two-factor solution for both datasets. Thereafter, we computed a five-factor solution (in accordance with the five domains present in the GAGE data). After computing the initial two- and five-factor solutions, we further refined the five-factor solution for both datasets. In particular, we excluded variables that failed to load onto a factor or exhibited cross-loadings. We iteratively refined the model until all remaining variables loaded onto a factor. We then applied cutoffs, removing variables with communalities below 0.2 [35,36] and factor loadings below 0.3 [34,35]. We repeated this process until all variables loaded onto a factor and did not produce cross-loadings.

## 3 Results

### 3.1 Sample characteristics

The Ethiopian sample contained 6,985 and the Bangladeshi sample 2,576 adolescents. After restricting the sample to those adolescents who answered all gender norm questions, the datasets comprise 6,183 individuals for Ethiopia (2,767 boys; 3,416 girls) and 2,245 individuals for Bangladesh (1,104 boys; 1,141 girls). The descriptive statistics of the gender norms items in the two datasets are presented in Table 1.

### 3.2 Exploratory Factor Analysis

Bartlett's test of sphericity yielded values of $\chi^2(435) = 46514.04$, $p < 0.001$ for the Ethiopia dataset and $\chi^2(435) = 13177.59$, $p < 0.001$ for the Bangladesh dataset, indicating that the inter-item correlations were sufficiently large for EFA in both datasets [34].

The overall KMO of the Ethiopia and Bangladesh datasets was 0.77 and 0.74, respectively, while the KMO of the individual variables ranged from 0.52 to 0.92 and from 0.55 to 0.86, respectively. With values above 0.5, the KMOs suggest adequate samples for conducting an EFA [34]. In the following, we describe the initial two- and five-factor solutions as well as the factor models of the refined five-factor solutions for Ethiopia and Bangladesh and compare the results between datasets.

**3.2.1 Two-factor solution.** The two-factor EFA for the Ethiopia dataset is shown in Table 2. Of the 30 variables, 14 loaded onto the first factor (threshold 0.30), while only three variables loaded onto the second factor. Of the former, four variables loaded strongly onto the first factor (threshold 0.50), while all three of the latter loaded strongly onto the second factor. Thirteen variables loaded onto neither factor.

**Table 1. Descriptive statistics of the GAGE gender norms items in the Ethiopia and Bangladesh datasets.**

| Gender norms | Sample distribution | |
|---|---|---|
| | Ethiopia n (%) | Bangladesh n (%) |
| **Education domain** | | |
| **cr_edu_boysch:** "If a family can afford for one child to go to secondary school it should be the boy only" | | |
| Agree | 1339 (21.7) | 383 (17.1) |
| Partially agree | 669 (10.8) | 269 (12.0) |
| Do not agree | 4175 (67.5) | 1593 (71.0) |
| **cr_edu_science:** "Only boys should learn about science, technology, and math" | | |
| Agree | 767 (12.4) | 192 (8.6) |
| Partially agree | 493 (8.0) | 200 (8.9) |
| Do not agree | 4923 (79.6) | 1853 (82.5) |
| **cr_edu_girlnohelp:** "Girls should be sent to school only if they are not needed to help at home" | | |
| Agree | 942 (15.2) | 267 (11.9) |
| Partially agree | 688 (11.1) | 251 (11.2) |
| Do not agree | 4553 (73.6) | 1727 (76.9) |
| **cr_edu_raisingvoice:** "Girls should avoid raising their voice to be lady like" | | |
| Agree | 2540 (41.1) | 1474 (65.7) |
| Partially agree | 967 (15.6) | 222 (9.9) |
| Do not agree | 2676 (43.3) | 549 (24.5) |
| **cr_edu_boysfeelings:** "Boys should be able to show their feelings without fear of being teased" | | |
| Agree | 4834 (78.2) | 666 (29.7) |
| Partially agree | 692 (11.2) | 338 (15.1) |
| Do not agree | 657 (10.6) | 1241 (55.3) |
| **cr_edu_culture:** "Our culture makes it harder for girls to achieve their goals than boys" | | |
| Agree | 3061 (49.5) | 1302 (58.0) |
| Partially agree | 770 (12.5) | 382 (17.0) |
| Do not agree | 2352 (38.0) | 561 (25.0) |
| **cr_edu_eegirlsout:** "Adolescent girls in my community are more likely to be out of school than adolescent boys" | | |
| Agree | 3187 (51.5) | 488 (21.7) |
| Partially agree | 1105 (17.9) | 445 (19.8) |
| Do not agree | 1891 (30.6) | 1312 (58.4) |
| **cr_edu_girlschoolsent:** "Girls in my community are sent to school only if they are not needed to help at home" | | |
| Agree | 1302 (21.1) | 332 (14.8) |
| Partially agree | 984 (15.9) | 330 (14.7) |
| Do not agree | 1897 (63.0) | 1583 (70.5) |
| **cr_edu_girlschoolexpect:** "Most people in my community expect girls to be sent to school only if they are not needed at home" | | |
| Agree | 1777 (28.7) | 361 (16.1) |
| Partially agree | 1123 (18.2) | 354 (15.8) |
| Do not agree | 3283 (53.1) | 1530 (68.2) |
| **Domain time use** | | |
| **cr_tu_statements1:** "Girls and boys should share household tasks equally" | | |
| Agree | 4212 (68.1) | 1871 (83.3) |
| Partially agree | 778 (12.6) | 174 (7.8) |

*(Continued)*

**Table 1.** (Continued)

| Gender norms | Sample distribution | |
|---|---|---|
| | Ethiopia n (%) | Bangladesh n (%) |
| Do not agree | 1193 (19.3) | 200 (08.9) |
| **cr_tu_statements2:** "A woman's most important role is to take care of her home and cook for her family" | | |
| Agree | 3681 (59.5) | 1147 (51.1) |
| Partially agree | 926 (15.0) | 325 (14.5) |
| Do not agree | 1576 (25.5) | 773 (34.4) |
| **cr_tu_statements3:** "A man should have the final word on decisions in his home" | | |
| Agree | 3537 (57.2) | 937 (41.7) |
| Partially agree | 972 (15.7) | 326 (14.5) |
| Do not agree | 1674 (27.1) | 982 (43.7) |
| **cr_tu_statements4:** "Most boys and girls in my community do not share household tasks equally" | | |
| Agree | 3871 (62.6) | 1356 (60.4) |
| Partially agree | 1005 (16.3) | 361 (16.1) |
| Do not agree | 1307 (21.1) | 528 (23.5) |
| **cr_tu_statements5:** "Most people in my community expect men to have the final word about decisions in the home" | | |
| Agree | 4085 (66.1) | 977 (43.5) |
| Partially agree | 1022 (16.5) | 454 (20.2) |
| Do not agree | 1076 (17.4) | 814 (36.3) |
| **cr_tu_statements6:** "Most people in my community do not expect girls and boys to share household tasks equally" | | |
| Agree | 4169 (67.4) | 1264 (56.3) |
| Partially agree | 979 (15.8) | 428 (19.1) |
| Do not agree | 1035 (16.7) | 553 (24.6) |
| **cr_tu_statements7:** "Most men in my community are the ones who make the decisions in their home" | | |
| Agree | 4085 (66.1) | 1091 (48.6) |
| Partially agree | 1041 (16.8) | 501 (22.3) |
| Do not agree | 1057 (17.1) | 653 (29.1) |
| **Domain financial inclusion and economic empowerment norms** | | |
| **cr_fin_girlchance:** "Women should have the same chance to work outside of the home as men" | | |
| Agree | 5106 (82.6) | 1850 (82.4) |
| Partially agree | 577 (9.3) | 166 (7.4) |
| Do not agree | 500 (8.1) | 229 (10.2) |
| **cr_fin_notgood:** "Women who participate in politics or leadership positions cannot also be good wives or mothers" | | |
| Agree | 1696 (27.4) | 653 (29.1) |
| Partially agree | 795 (12.9) | 426 (19.0) |
| Do not agree | 3692 (59.7) | 1166 (51.9) |
| **cr_fin_impwsav:** "It is important for women and adolescent girls to have their own savings" | | |
| Agree | 5521 (89.3) | 1999 (89.0) |
| Partially agree | 470 (7.6) | 165 (7.4) |
| Do not agree | 192 (3.1) | 81 (3.6) |
| **cr_fin_eework:** "Most women in my community have the same chance to work outside the home as men" | | |
| Agree | 3822 (61.8) | 1400 (62.4) |
| Partially agree | 1188 (19.2) | 443 (19.7) |
| Do not agree | 1173 (19.0) | 402 (17.9) |

*(Continued)*

**Table 1.** (Continued)

| Gender norms | Sample distribution | |
|---|---|---|
| | Ethiopia n (%) | Bangladesh n (%) |
| **cr_fin_nework:** "Most people in my community expect women to have the same chance to work outside the home as men" | | |
| Agree | 3606 (58.3) | 1432 (63.8) |
| Partially agree | 1324 (21.4) | 458 (20.4) |
| Do not agree | 1253 (20.3) | 355 (15.8) |
| **Domain marriage and relationships norms** | | |
| **cr_mar_girlfriend:** "A boy should be able to have a girlfriend if he wants to" | | |
| Agree | 3850 (62.3) | 1869 (83.5) |
| Partially agree | 669 (10.8) | 153 (6.8) |
| Do not agree | 1664 (26.9) | 223 (9.9) |
| **cr_mar_boyfriend:** "A girl should be able to have a boyfriend if she wants to" | | |
| Agree | 3487 (56.4) | 1821 (81.1) |
| Partially agree | 763 (12.3) | 180 (8.0) |
| Do not agree | 1933 (31.3) | 244 (10.9) |
| **cr_mar_waitedu:** "A girl's marriage can wait until she has completed secondary school" | | |
| Agree | 4719 (76.3) | 1712 (76.3) |
| Partially agree | 698 (11.3) | 180 (8.0) |
| Do not agree | 766 (12.4) | 353 (15.7) |
| **cr_mar_eemarryage:** "Most adolescent girls in my community marry before the age of 18 years" (legal age) | | |
| Agree | 2534 (41.0) | 841 (37.5) |
| Partially agree | 944 (15.3) | 442 (19.7) |
| Do not agree | 2705 (43.8) | 962 (42.9) |
| **cr_mar_nemarryage:** "Adults in my community expect adolescent girls to get married before the age of 18 years" (legal age) | | |
| Agree | 2325 (37.6) | 534 (23.8) |
| Partially agree | 939 (15.2) | 438 (19.5) |
| Do not agree | 2919 (47.2) | 1273 (56.7) |
| **Domain sexual and reproductive health norms** | | |
| **cr_srh_proudbod:** "Girls should be proud of their bodies as they become women" | | |
| Agree | 4771 (77.2) | 1236 (55.1) |
| Partially agree | 523 (8.5) | 344 (15.3) |
| Do not agree | 889 (14.4) | 665 (29.6) |
| **cr_srh_controldaught:** "Families should control their daughters' behaviors more than their sons" | | |
| Agree | 4123 (66.7) | 1392 (62.0) |
| Partially agree | 805 (13.0) | 395 (17.6) |
| Do not agree | 1255 (20.3) | 458 (20.4) |
| **cr_srh_eecontrolgirls:** "Most families in my community control their daughters' behaviors more than their sons'" | | |
| Agree | 4613 (74.6) | 1454 (64.8) |
| Partially agree | 818 (13.2) | 421 (18.8) |
| Do not agree | 752 (12.2) | 370 (16.5) |
| **cr_srh_necontrolgirls:** "Most people in my community expect families to control their daughter's behavior more than their sons'" | | |
| Agree | 4558 (73.7) | 1404 (62.5) |
| Partially agree | 857 (13.9) | 462 (20.6) |
| Do not agree | 768 (12.4) | 379 (16.9) |

**Table 2. Two-factor solution for the Ethiopia dataset (N = 6,183).**

| Question content | Factor 1 | Factor 2 | Communality | Uniqueness |
|---|---|---|---|---|
| If a family can afford for one child to go to secondary school it should be the boy only | 0.52 | -0.02 | 0.27 | 0.73 |
| Only boys should learn about science, technology, and math | 0.51 | -0.07 | 0.25 | 0.75 |
| Girls should be sent to school only if they are not needed to help at home | 0.54 | -0.09 | 0.28 | 0.72 |
| Girls should avoid raising their voice to be lady like | 0.39 | 0.05 | 0.16 | 0.84 |
| Boys should be able to show their feelings without fear of being teased | 0.03 | -0.11 | 0.01 | 1.00 |
| Girls and boys should share household tasks equally | 0.23 | -0.14 | 0.06 | 0.94 |
| A woman's most important role is to take care of her home and cook for her family | 0.43 | 0.13 | 0.23 | 0.77 |
| A man should have the final word on decisions in his home | 0.46 | 0.20 | 0.29 | 0.71 |
| Women should have the same chance to work outside of the home as men | 0.26 | -0.27 | 0.11 | 0.89 |
| Women who participate in politics or leadership positions cannot also be good wives or mothers | 0.39 | 0.07 | 0.16 | 0.84 |
| It is important for women and adolescent girls to have their own savings | 0.23 | -0.27 | 0.10 | 0.90 |
| A boy should be able to have a girlfriend if he wants to | 0.01 | -0.21 | 0.04 | 0.96 |
| A girl should be able to have a boyfriend if she wants to | 0.08 | -0.21 | 0.04 | 0.96 |
| A girl's marriage can wait until she has completed secondary school | 0.10 | -0.17 | 0.03 | 0.97 |
| Girls should be proud of their bodies as they become women | 0.00 | -0.16 | 0.03 | 0.97 |
| Families should control their daughters' behaviors more than their sons | 0.10 | 0.54 | 0.33 | 0.67 |
| Our culture makes it harder for girls to achieve their goals than boys | 0.45 | 0.02 | 0.21 | 0.79 |
| Adolescent girls in my community are more likely to be out of school than adolescent boys | 0.30 | 0.08 | 0.10 | 0.90 |
| Girls in my community are sent to school only if they are not needed to help at home | 0.58 | -0.09 | 0.32 | 0.68 |
| Most people in my community expect girls to be sent to school only if they are not needed at home | 0.48 | -0.04 | 0.22 | 0.78 |
| Most boys and girls in my community do not share household tasks equally | 0.19 | 0.20 | 0.09 | 0.91 |
| Most people in my community expect men to have the final word about decisions in the home | 0.39 | 0.24 | 0.25 | 0.75 |
| Most people in my community do not expect girls and boys to share household tasks equally | 0.28 | 0.22 | 0.15 | 0.85 |
| Most men in my community are the ones who make the decisions in their home | 0.37 | 0.26 | 0.25 | 0.75 |
| Most women in my community have the same chance to work outside the home as men | 0.29 | -0.24 | 0.11 | 0.89 |
| Most people in my community expect women to have the same chance to work outside the home as men | 0.28 | -0.22 | 0.10 | 0.90 |
| Most adolescent girls in my community marry before the age of 18 years (legal age) | 0.39 | 0.07 | 0.17 | 0.83 |
| Adults in my community expect adolescent girls to get married before the age of 18 years (legal age) | 0.40 | 0.06 | 0.18 | 0.82 |
| Most families in my community control their daughters' behaviors more than their sons' | -0.01 | 0.72 | 0.51 | 0.49 |
| Most people in my community expect families to control their daughter's behavior more than their sons' | 0.00 | 0.71 | 0.50 | 0.50 |
| Cumulative variance explained | 11.62% | 18.55% | | |

Communality and uniqueness are rounded to the second decimal place

Green highlighted factor loadings = above 0.50 (strong loading)

Yellow highlighted factor loadings = above 0.30 (weak loading)

Grey highlighted rows = community-level gender norms

Notably, the loading patterns do not correspond to the expected two factors (perceived community-level gender norms and individual-level gender attitudes). In particular, the second factor contains only variables on what was defined as *sexual and reproductive health*, including two community-level variables and one individual-level variable. The first factor also contains both individual- and community-level variables, particularly on *education* and *time use*.

Furthermore, the two-factor solution does not fit the data well. Beyond the high number of variables that do not load onto any factor, the communalities of the variables are low across the board. In fact, only two variables have a communality of more than 0.50. A substantial number of variables has a very high uniqueness (higher than 0.80), indicating that they

contribute little to the factor structure. Only two variables have uniqueness values below 0.60. The two factors explained 11.62% and 6.93% of the variance, respectively, for a total of only 18.55%.

In the Bangladesh dataset, seven variables loaded onto the first factor, three of them strongly. Another seven variables loaded onto the second factor, only a single one of them strongly. The remaining sixteen variables loaded onto neither factor. The first factor contains most of the *education* variables, including some (but not all) individual- and community-level variables. The second factor contains most of the variables (both individual- and community-level) on *sexual and reproductive health*, along with some *time-use* variables and one *education* variable. The two-factor solution for the Bangladesh dataset is shown in Table 3.

Once again, therefore, the loadings do not correspond well to the individual-community distinction. Furthermore, as in the Ethiopia dataset, the factor solution does not adequately fit the data. Communalities are again low, with no variable having a communality above 0.43. The cumulative proportion of explained variance is only 15.48%, with 8.53% of variance explained by the first and 6.96% by the second factor.

In summary, the two-factor solution is unsatisfactory in both datasets. This indicates that it is not a reflection of the data structure to divide the variables into a community-level and an individual-level factor. Instead, the fact that both datasets yield a factor that contains only or mostly variables from a single domain indicates that a five-factor solution based on these domains may be more appropriate.

**3.2.2  Initial five-factor solution.**  The five-factor solution for the Ethiopia dataset, shown in Table 4, generally supports this notion. Each of the five factors corresponds roughly to one of the five domains. The first factor is strongly loaded onto five out of nine *education* variables, with another *education* variable loading weakly. In contrast, only one non-*education* variable loads (weakly) onto the first factor. The second factor contains all but one of the *time-use* variables, with most of these variables loading strongly. Two non-*time-use* variables also loaded onto the second factor, though they only loaded weakly. Three of four *sexual and reproductive health* variables loaded strongly onto the third factor. The fourth factor includes mostly *financial and economic empowerment* variables, while the fifth factor includes only two variables, both of which belong to the *marriage and relationship* domain.

Although the communalities in this factor solution were somewhat higher than in the two-factor solution, a number of variables with very low communalities remain. Furthermore, although the number of unassigned variables is lower than in the two-factor solution, six variables remain that do not load onto any factor. There now is a higher number of variables with uniqueness values below 0.6, but 13 variables remain with values around 0.8 and higher. And while the cumulative proportion of variance explained is higher than for either two-factor solution at 33.82%, this proportion is still below the desired minimum of 50% in social sciences [37].

In the Bangladesh dataset, a similar picture emerges, with the five factors consisting mainly of *education*, *sexual and reproductive health*, *marriage and relationship*, *time-use*, and *financial and economic empowerment variables*, respectively. Ten variables did not load onto any factor. The five-factor solution for the Bangladesh dataset is shown in Table 5. A number of variables with very low communalities remain; 12 variables had a uniqueness above 0.8 and only eight variables had a uniqueness below 0.6. The cumulative proportion of variance explained was only 29.84%.

These results support the conclusion that dividing the variables by domain is more appropriate than dividing them into two factors with individual-level and community-level variables. However, the five-factor solutions remain far from perfect in both datasets, exhibiting low communalities, non-loading variables and low proportions of explained variance.

These issues could indicate that a number of factors other than two or five may provide a better factor solution. However, they may also reflect issues with the original data, and the need to reconsider the wording or domain assignment of some questions. Therefore, we next investigated whether removing problematic items improves the five-factor solutions in the Ethiopia and Bangladesh datasets.

**3.2.3  Refined five-factor solution.**  To refine the initial five-factor solution for Ethiopia, we removed six variables that failed to load onto any of the five factors (cr_fin_impwsav, cr_edu_raisingvoice, cr_mar_waitedu, cr_edu_eegirlsout,

**Table 3. Two-factor solution for the Bangladesh dataset (N = 2,245).**

| Question content | Factor 1 | Factor 2 | Communality | Uniqueness |
|---|---|---|---|---|
| If a family can afford for one child to go to secondary school it should be the boy only | 0.43 | 0.01 | 0.19 | 0.81 |
| Only boys should learn about science, technology, and math | 0.47 | 0.02 | 0.23 | 0.77 |
| Girls should be sent to school only if they are not needed to help at home | 0.63 | -0.06 | 0.38 | 0.62 |
| Girls should avoid raising their voice to be lady like | -0.18 | 0.37 | 0.15 | 0.85 |
| Boys should be able to show their feelings without fear of being teased | -0.23 | -0.17 | 0.10 | 0.90 |
| Girls and boys should share household tasks equally | 0.14 | -0.06 | 0.02 | 0.98 |
| A woman's most important role is to take care of her home and cook for her family | 0.27 | 0.35 | 0.23 | 0.77 |
| A man should have the final word on decisions in his home | 0.37 | 0.28 | 0.25 | 0.75 |
| Women should have the same chance to work outside of the home as men | 0.25 | 0.02 | 0.07 | 0.93 |
| Women who participate in politics or leadership positions cannot also be good wives or mothers | 0.13 | 0.17 | 0.05 | 0.95 |
| It is important for women and adolescent girls to have their own savings | 0.25 | -0.06 | 0.06 | 0.94 |
| A boy should be able to have a girlfriend if he wants to | 0.21 | -0.42 | 0.19 | 0.81 |
| A girl should be able to have a boyfriend if she wants to | 0.20 | -0.39 | 0.16 | 0.84 |
| A girl's marriage can wait until she has completed secondary school | 0.22 | -0.33 | 0.13 | 0.87 |
| Girls should be proud of their bodies as they become women | 0.01 | -0.26 | 0.06 | 0.94 |
| Families should control their daughters' behaviors more than their sons | 0.04 | 0.36 | 0.14 | 0.86 |
| Our culture makes it harder for girls to achieve their goals than boys | 0.12 | 0.14 | 0.04 | 0.96 |
| Adolescent girls in my community are more likely to be out of school than adolescent boys | 0.31 | -0.13 | 0.10 | 0.90 |
| Girls in my community are sent to school only if they are not needed to help at home | 0.66 | -0.01 | 0.43 | 0.57 |
| Most people in my community expect girls to be sent to school only if they are not needed at home | 0.57 | 0.04 | 0.33 | 0.67 |
| Most boys and girls in my community do not share household tasks equally | 0.00 | 0.22 | 0.05 | 0.95 |
| Most people in my community expect men to have the final word about decisions in the home | 0.27 | 0.37 | 0.24 | 0.76 |
| Most people in my community do not expect girls and boys to share household tasks equally | 0.01 | 0.11 | 0.01 | 0.99 |
| Most men in my community are the ones who make the decisions in their home | 0.18 | 0.43 | 0.24 | 0.76 |
| Most women in my community have the same chance to work outside the home as men | 0.15 | 0.12 | 0.04 | 0.96 |
| Most people in my community expect women to have the same chance to work outside the home as men | 0.15 | 0.10 | 0.04 | 0.96 |
| Most adolescent girls in my community marry before the age of 18 years (legal age) | 0.22 | 0.19 | 0.10 | 0.90 |
| Adults in my community expect adolescent girls to get married before the age of 18 years (legal age) | 0.24 | 0.18 | 0.11 | 0.89 |
| Most families in my community control their daughters' behaviors more than their sons' | 0.03 | 0.48 | 0.23 | 0.77 |
| Most people in my community expect families to control their daughter's behavior more than their sons' | 0.00 | 0.52 | 0.27 | 0.73 |
| Cumulative variance explained | 8.53% | 15.48% | | |

Communality and uniqueness are rounded to the second decimal place

Green highlighted factor loadings = above 0.50 (strong loading)

Yellow highlighted factor loadings = above 0.30 (weak loading)

Grey highlighted rows = community-level gender norms

cr_edu_boysfeelings, cr_srh_proudbod). The exclusion of these items slightly reduced the KMO from 0.77 to 0.75. Thereafter, we removed another three variables that did not load onto any factor in the reduced set of items (cr_tu_statements1, cr_mar_eemarryage, cr_mar_nemarryage). This resulted in all variables loading onto at least one of the five factors. Further exclusion of items did not result in any additional change in the KMO. Finally, we excluded four more variables due to low communalities (<0.2) and factor loadings (<0.3) (cr_fin_notgood, cr_fin_girlchance, cr_edu_culture, cr_tu_statements4), resulting in a KMO of 0.71. The final set of 17 variables is shown in Table 6.

**Table 4. Initial five-factor solution for the Ethiopia dataset (N = 6,183).**

| Question content | F1 | F2 | F3 | F4 | F5 | Communality | Uniqueness |
|---|---|---|---|---|---|---|---|
| If a family can afford for one child to go to secondary school it should be the boy only | 0.53 | 0.08 | 0.01 | 0.03 | -0.07 | 0.33 | 0.67 |
| Only boys should learn about science, technology, and math | 0.61 | -0.02 | 0.00 | 0.00 | -0.05 | 0.37 | 0.63 |
| Girls should be sent to school only if they are not needed to help at home | 0.69 | -0.04 | -0.01 | -0.04 | -0.02 | 0.46 | 0.54 |
| Girls should avoid raising their voice to be lady like | 0.25 | 0.22 | 0.01 | 0.06 | -0.04 | 0.16 | 0.84 |
| Boys should be able to show their feelings without fear of being teased | 0.07 | -0.14 | 0.01 | 0.10 | 0.06 | 0.03 | 0.97 |
| Girls and boys should share household tasks equally | 0.07 | 0.04 | -0.02 | 0.31 | 0.01 | 0.11 | 0.89 |
| A woman's most important role is to take care of her home and cook for her family | 0.17 | 0.45 | -0.02 | -0.01 | 0.02 | 0.28 | 0.72 |
| A man should have the final word on decisions in his home | 0.11 | 0.56 | -0.01 | 0.01 | -0.02 | 0.38 | 0.62 |
| Women should have the same chance to work outside of the home as men | 0.13 | -0.08 | -0.05 | 0.43 | 0.03 | 0.20 | 0.80 |
| Women who participate in politics or leadership positions cannot also be good wives or mothers | 0.36 | 0.12 | 0.05 | -0.02 | -0.05 | 0.19 | 0.81 |
| It is important for women and adolescent girls to have their own savings | 0.26 | -0.18 | -0.06 | 0.23 | 0.05 | 0.13 | 0.87 |
| A boy should be able to have a girlfriend if he wants to | -0.02 | -0.02 | 0.00 | 0.00 | 0.88 | 0.78 | 0.22 |
| A girl should be able to have a boyfriend if she wants to | 0.03 | 0.03 | 0.00 | 0.01 | 0.88 | 0.78 | 0.22 |
| A girl's marriage can wait until she has completed secondary school | 0.19 | -0.16 | -0.04 | 0.07 | 0.05 | 0.05 | 0.95 |
| Girls should be proud of their bodies as they become women | -0.03 | -0.01 | -0.11 | 0.04 | 0.12 | 0.03 | 0.97 |
| Families should control their daughters' behaviors more than their sons | 0.10 | 0.05 | 0.57 | 0.00 | -0.07 | 0.38 | 0.62 |
| Our culture makes it harder for girls to achieve their goals than boys | 0.32 | 0.17 | 0.04 | 0.12 | -0.02 | 0.20 | 0.80 |
| Adolescent girls in my community are more likely to be out of school than adolescent boys | 0.17 | 0.17 | 0.08 | 0.07 | 0.03 | 0.10 | 0.90 |
| Girls in my community are sent to school only if they are not needed to help at home | 0.66 | 0.01 | 0.01 | 0.00 | 0.07 | 0.44 | 0.56 |
| Most people in my community expect girls to be sent to school only if they are not needed at home | 0.55 | 0.03 | 0.03 | -0.05 | 0.07 | 0.32 | 0.68 |
| Most boys and girls in my community do not share household tasks equally | -0.03 | 0.33 | 0.07 | 0.01 | -0.03 | 0.13 | 0.87 |
| Most people in my community expect men to have the final word about decisions in the home | -0.05 | 0.68 | 0.00 | 0.02 | 0.01 | 0.45 | 0.55 |
| Most people in my community do not expect girls and boys to share household tasks equally | 0.00 | 0.50 | 0.00 | -0.09 | 0.02 | 0.25 | 0.75 |
| Most men in my community are the ones who make the decisions in their home | -0.05 | 0.67 | 0.02 | 0.00 | 0.01 | 0.43 | 0.57 |
| Most women in my community have the same chance to work outside the home as men | -0.02 | 0.00 | 0.00 | 0.80 | -0.01 | 0.64 | 0.36 |
| Most people in my community expect women to have the same chance to work outside the home as men | -0.03 | 0.02 | 0.02 | 0.75 | 0.01 | 0.56 | 0.44 |
| Most adolescent girls in my community marry before the age of 18 years (legal age) | 0.12 | 0.31 | 0.03 | 0.19 | -0.04 | 0.20 | 0.80 |
| Adults in my community expect adolescent girls to get married before the age of 18 years (legal age) | 0.14 | 0.31 | 0.02 | 0.16 | -0.04 | 0.19 | 0.81 |
| Most families in my community control their daughters' behaviors more than their sons' | 0.01 | -0.03 | 0.92 | 0.00 | 0.00 | 0.82 | 0.18 |
| Most people in my community expect families to control their daughter's behavior more than their sons' | -0.01 | 0.01 | 0.86 | 0.00 | 0.02 | 0.75 | 0.25 |
| Cumulative variance explained | 8.44% | 16.24% | 22.84% | 28.42% | 33.82% | | |

Communality and uniqueness are rounded to the second decimal place

Green highlighted factor loadings = above 0.50 (strong loading)

Yellow highlighted factor loadings = above 0.30 (weak loading)

Grey highlighted rows = community-level gender norms

**Table 5. Initial five-factor solution for the Bangladesh dataset (N = 2,245).**

| Question content | F1 | F2 | F3 | F4 | F5 | Communality | Uniqueness |
|---|---|---|---|---|---|---|---|
| If a family can afford for one child to go to secondary school it should be the boy only | 0.35 | -0.05 | 0.05 | 0.19 | -0.01 | 0.19 | 0.81 |
| Only boys should learn about science, technology, and math | 0.40 | -0.06 | 0.02 | 0.19 | -0.03 | 0.24 | 0.76 |
| Girls should be sent to school only if they are not needed to help at home | 0.61 | -0.06 | 0.02 | 0.07 | 0.02 | 0.41 | 0.59 |
| Girls should avoid raising their voice to be lady like | -0.23 | 0.11 | -0.18 | 0.23 | 0.03 | 0.14 | 0.86 |
| Boys should be able to show their feelings without fear of being teased | -0.19 | -0.03 | 0.06 | -0.21 | 0.06 | 0.11 | 0.89 |
| Girls and boys should share household tasks equally | 0.10 | -0.01 | 0.06 | -0.04 | 0.15 | 0.04 | 0.96 |
| A woman's most important role is to take care of her home and cook for her family | 0.04 | 0.00 | -0.04 | 0.54 | 0.07 | 0.33 | 0.67 |
| A man should have the final word on decisions in his home | 0.13 | -0.02 | 0.04 | 0.58 | -0.02 | 0.38 | 0.62 |
| Women should have the same chance to work outside of the home as men | 0.08 | -0.04 | 0.13 | 0.11 | 0.41 | 0.22 | 0.78 |
| Women who participate in politics or leadership positions cannot also be good wives or mothers | 0.03 | 0.01 | -0.03 | 0.23 | 0.03 | 0.07 | 0.93 |
| It is important for women and adolescent girls to have their own savings | 0.15 | -0.10 | 0.07 | 0.08 | 0.17 | 0.08 | 0.92 |
| A boy should be able to have a girlfriend if he wants to | 0.00 | 0.00 | 0.89 | -0.01 | 0.01 | 0.80 | 0.20 |
| A girl should be able to have a boyfriend if she wants to | -0.02 | 0.02 | 0.88 | 0.02 | 0.00 | 0.76 | 0.24 |
| A girl's marriage can wait until she has completed secondary school | 0.26 | -0.05 | 0.20 | -0.23 | 0.01 | 0.14 | 0.86 |
| Girls should be proud of their bodies as they become women | 0.01 | -0.04 | 0.19 | -0.16 | 0.01 | 0.08 | 0.92 |
| Families should control their daughters' behaviors more than their sons | 0.04 | 0.60 | 0.05 | 0.00 | -0.06 | 0.36 | 0.64 |
| Our culture makes it harder for girls to achieve their goals than boys | 0.14 | 0.20 | 0.01 | 0.01 | -0.06 | 0.07 | 0.93 |
| Adolescent girls in my community are more likely to be out of school than adolescent boys | 0.39 | 0.13 | 0.14 | -0.18 | -0.07 | 0.18 | 0.82 |
| Girls in my community are sent to school only if they are not needed to help at home | 0.69 | -0.01 | -0.02 | 0.03 | 0.03 | 0.49 | 0.51 |
| Most people in my community expect girls to be sent to school only if they are not needed at home | 0.61 | 0.01 | -0.07 | 0.05 | 0.00 | 0.40 | 0.60 |
| Most boys and girls in my community do not share household tasks equally | -0.01 | 0.28 | 0.04 | 0.04 | -0.02 | 0.08 | 0.92 |
| Most people in my community expect men to have the final word about decisions in the home | 0.02 | 0.05 | 0.02 | 0.57 | 0.03 | 0.36 | 0.64 |
| Most people in my community do not expect girls and boys to share household tasks equally | 0.01 | 0.15 | 0.03 | 0.06 | -0.12 | 0.04 | 0.96 |
| Most men in my community are the ones who make the decisions in their home | -0.03 | 0.12 | -0.02 | 0.54 | -0.04 | 0.33 | 0.67 |
| Most women in my community have the same chance to work outside the home as men | 0.01 | 0.03 | -0.02 | -0.02 | 0.71 | 0.50 | 0.50 |
| Most people in my community expect women to have the same chance to work outside the home as men | -0.01 | -0.01 | 0.00 | 0.00 | 0.76 | 0.57 | 0.43 |
| Most adolescent girls in my community marry before the age of 18 years (legal age) | 0.33 | 0.30 | -0.09 | -0.11 | -0.01 | 0.19 | 0.81 |
| Adults in my community expect adolescent girls to get married before the age of 18 years (legal age) | 0.34 | 0.24 | -0.11 | -0.09 | 0.03 | 0.17 | 0.83 |
| Most families in my community control their daughters' behaviors more than their sons' | -0.01 | 0.78 | 0.03 | 0.01 | 0.02 | 0.62 | 0.38 |
| Most people in my community expect families to control their daughter's behavior more than their sons' | -0.03 | 0.75 | -0.03 | 0.04 | 0.02 | 0.58 | 0.42 |
| Cumulative variance explained | 7.26% | 13.64% | 19.57% | 25.34% | 29.84% | | |

Communality and uniqueness are rounded to the second decimal place

Green highlighted factor loadings = above 0.50 (strong loading)

Yellow highlighted factor loadings = above 0.30 (weak loading)

Grey highlighted rows = community-level gender norms

**Table 6. Refined five-factor solution for the Ethiopia dataset (N = 6,183).**

| Question content | F1 | F2 | F3 | F4 | F5 | Communality | Uniqueness |
|---|---|---|---|---|---|---|---|
| If a family can afford for one child to go to secondary school it should be the boy only | 0.50 | 0.01 | 0.08 | -0.07 | 0.04 | 0.29 | 0.71 |
| Only boys should learn about science, technology, and math | 0.58 | -0.01 | 0.01 | 0.06 | 0.01 | 0.35 | 0.65 |
| Girls should be sent to school only if they are not needed to help at home | 0.71 | -0.02 | -0.03 | -0.03 | -0.02 | 0.49 | 0.51 |
| A woman's most important role is to take care of her home and cook for her family | 0.17 | -0.01 | 0.45 | 0.02 | -0.03 | 0.28 | 0.72 |
| A man should have the final word on decisions in his home | 0.09 | 0.00 | 0.62 | -0.02 | -0.02 | 0.42 | 0.58 |
| A boy should be able to have a girlfriend if he wants to | -0.02 | 0.00 | -0.02 | 0.89 | 0.00 | 0.80 | 0.20 |
| A girl should be able to have a boyfriend if she wants to | 0.02 | 0.00 | 0.03 | 0.88 | 0.00 | 0.77 | 0.23 |
| Families should control their daughters' behaviors more than their sons | 0.07 | 0.57 | 0.06 | -0.07 | -0.04 | 0.38 | 0.62 |
| Girls in my community are sent to school only if they are not needed to help at home | 0.70 | 0.01 | -0.01 | 0.06 | 0.03 | 0.49 | 0.51 |
| Most people in my community expect girls to be sent to school only if they are not needed at home | 0.57 | 0.03 | 0.02 | 0.06 | -0.03 | 0.34 | 0.66 |
| Most people in my community expect men to have the final word about decisions in the home | -0.05 | 0.00 | 0.73 | 0.01 | 0.03 | 0.51 | 0.49 |
| Most people in my community do not expect girls and boys to share household tasks equally | 0.04 | 0.01 | 0.47 | 0.02 | -0.05 | 0.23 | 0.77 |
| Most men in my community are the ones who make the decisions in their home | -0.05 | 0.02 | 0.69 | 0.00 | 0.01 | 0.47 | 0.53 |
| Most women in my community have the same chance to work outside the home as men | 0.01 | -0.01 | 0.00 | -0.01 | 0.81 | 0.65 | 0.35 |
| Most people in my community expect women to have the same chance to work outside the home as men | 0.00 | 0.01 | 0.00 | 0.01 | 0.81 | 0.66 | 0.34 |
| Most families in my community control their daughters' behaviors more than their sons' | -0.01 | 0.91 | -0.03 | 0.00 | 0.01 | 0.82 | 0.18 |
| Most people in my community expect families to control their daughter's behavior more than their sons' | -0.01 | 0.86 | 0.01 | 0.01 | 0.00 | 0.75 | 0.25 |
| Cumulative variance explained | 11.74% | 23.05% | 34.03% | 43.38% | 51.12% | | |

Communality and uniqueness are rounded to the second decimal place

Green highlighted factor loadings = above 0.50 (strong loading)

Yellow highlighted factor loadings = above 0.30 (weak loading)

Grey highlighted rows = community-level gender norms

The refined five-factor solution shows that each factor corresponds to exactly one of the five domains. The first factor contains five *education* variables with four of them loading strongly onto the factor. The second factor is strongly loaded onto by three *sexual and reproductive health* variables while the third factor is loaded onto by five *time-use* variables, with most of them loading strongly. The fourth and fifth factors only include two variables each. The two *marriage and relationship* variables loaded strongly onto the fourth and the two *financial and economic empowerment* variables loaded strongly onto the fifth factor. As a result, each of the five domains in the GAGE gender norms scale loads onto one of the five factors.

Although some communalities remain relatively low in the refined five-factor solution, the lowest value is now 0.23. While six variables have uniqueness values higher than 0.6, none exceed 0.8. The cumulative proportion of variance explained with over 51% is noticeably higher than in the initial five-factor solution.

The results in the Bangladesh dataset, displayed in Table 7, show a similar picture. After removing ten variables that failed to load onto any of the five factors (cr_edu_raisingvoice, cr_edu_boysfeelings, cr_tu_statements1, cr_fin_notgood,

**Table 7. Refined five-factor solution for the Bangladesh dataset (2,245).**

| Question content | F1 | F2 | F3 | F4 | F5 | Communality | Uniqueness |
|---|---|---|---|---|---|---|---|
| Only boys should learn about science, technology, and math | -0.01 | 0.40 | 0.01 | 0.14 | -0.04 | 0.22 | 0.78 |
| Girls should be sent to school only if they are not needed to help at home | -0.03 | 0.66 | 0.03 | 0.02 | 0.01 | 0.45 | 0.55 |
| A woman's most important role is to take care of her home and cook for her family | 0.00 | 0.05 | -0.05 | 0.50 | 0.05 | 0.29 | 0.71 |
| A man should have the final word on decisions in his home | -0.02 | 0.10 | 0.02 | 0.59 | -0.04 | 0.38 | 0.62 |
| A boy should be able to have a girlfriend if he wants to | -0.01 | 0.01 | 0.90 | -0.02 | 0.01 | 0.81 | 0.19 |
| A girl should be able to have a boyfriend if she wants to | 0.01 | -0.01 | 0.90 | 0.02 | -0.01 | 0.81 | 0.19 |
| Families should control their daughters' behaviors more than their sons | 0.61 | 0.02 | 0.04 | 0.02 | -0.08 | 0.38 | 0.62 |
| Girls in my community are sent to school only if they are not needed to help at home | 0.02 | 0.78 | 0.00 | -0.03 | 0.01 | 0.58 | 0.42 |
| Most people in my community expect girls to be sent to school only if they are not needed at home | 0.01 | 0.62 | -0.04 | 0.03 | 0.00 | 0.41 | 0.59 |
| Most people in my community expect men to have the final word about decisions in the home | -0.01 | -0.02 | 0.02 | 0.66 | 0.04 | 0.43 | 0.57 |
| Most men in my community are the ones who make the decisions in their home | 0.06 | -0.07 | -0.02 | 0.61 | -0.02 | 0.36 | 0.64 |
| Most women in my community have the same chance to work outside the home as men | 0.01 | 0.00 | -0.01 | -0.01 | 0.75 | 0.56 | 0.44 |
| Most people in my community expect women to have the same chance to work outside the home as men | -0.01 | 0.00 | 0.01 | 0.01 | 0.76 | 0.57 | 0.43 |
| Most families in my community control their daughters' behaviors more than their sons' | 0.84 | 0.00 | 0.02 | -0.01 | 0.01 | 0.71 | 0.29 |
| Most people in my community expect families to control their daughter's behavior more than their sons' | 0.79 | 0.00 | -0.04 | 0.01 | 0.02 | 0.63 | 0.37 |
| Cumulative variance explained | 11.41% | 22.32% | 33.22% | 42.93% | 50.57% | | |

Communality and uniqueness are rounded to the second decimal place

Green highlighted factor loadings = above 0.50 (strong loading)

Yellow highlighted factor loadings = above 0.30 (weak loading)

Grey highlighted rows = community-level gender norms

cr_fin_impwsav, cr_mar_waitedu, cr_srh_proudbod, cr_edu_culture, cr_tu_statements4, cr_tu_statements6), all variables loaded onto one of the five factors. Thereafter, we further excluded five variables with low communalities and factor loadings (cr_edu_boysch, cr_fin_girlchance, cr_edu_eegirlsout, cr_mar_eemarryage, cr_mar_nemarryage). This process resulted in a final set of 15 variables. The KMO decreased from 0.74 and 0.72 to 0.69, which remains an acceptable, but not ideal value to conduct EFAs.

The refined five-factor solution contains three *sexual and reproductive health* variables that loaded strongly onto the first factor. All four *education* variables loaded onto the second factor (three of them strongly), while two *marriage and relationship* variables loaded strongly onto the third factor. Four *time-use* variables loaded onto the fourth factor, with three of them loading strongly, while *two financial and economic empowerment* variables loaded strongly onto the fifth factor. Similarly to Ethiopia, each of the five domains in the GAGE gender norms scale loads onto one of the five factors.

The communalities and uniqueness values are comparable to the Ethiopia factor solution. No very low communalities and no uniqueness values higher than 0.8 remain. Only five uniqueness values are slightly above 0.6. As in the Ethiopia dataset, the cumulative variance accounts for an acceptable proportion of 50.57%.

Comparing the refined five-factor solutions in Table 8, each factor corresponds exactly to one of the five domains. With the exception of the two additional variables in the Ethiopia dataset and one variable that loads weaker in the Bangladesh dataset, the same variables load with the same strength onto the factors. Table 8 also shows that individual-level gender attitudes and community-level gender norms are mixed within the domains (at least where a sufficient number of variables is available per factor). This further supports the notion that the individual-community distinction is not suitable for the available GAGE gender norms items in Ethiopia and Bangladesh.

## 4  Discussion

### 4.1  Principal findings

To assess whether the items and domains of the GAGE gender norms scale contribute to explaining gender norms and attitudes in Ethiopia and Bangladesh, we explored the implied two- and five-factor structures (both in the GAGE gender norms scale and in Baird et al.'s analysis [29]) using EFA. The two-factor structure, distinguishing individual-level gender attitudes and community-level gender norms, showed an unsatisfactory model fit in both datasets due to the pattern of factor loadings not reflecting the individual-community distinction as well as low communalities, high uniqueness values

**Table 8.  Comparison of the refined five-factor structure with variables loading significantly onto a factor in the Ethiopia and the Bangladesh datasets.**

| Question content | Ethiopia | | | | | Bangladesh | | | | |
|---|---|---|---|---|---|---|---|---|---|---|
| | F1 | F2 | F3 | F4 | F5 | F1 | F2 | F3 | F4 | F5 |
| **Education domain** | | | | | | | | | | |
| If a family can afford for one child to go to secondary school it should be the boy only | ○ | | | | | | | | | |
| Only boys should learn about science, technology, and math | ● | | | | | | ○ | | | |
| Girls should be sent to school only if they are not needed to help at home | ● | | | | | | ● | | | |
| Girls in my community are sent to school only if they are not needed to help at home | ● | | | | | | ● | | | |
| Most people in my community expect girls to be sent to school only if they are not needed at home | ● | | | | | | ● | | | |
| **Time use domain** | | | | | | | | | | |
| A woman's most important role is to take care of her home and cook for her family | | | ○ | | | | | | ○ | |
| A man should have the final word on decisions in his home | | | ● | | | | | | ● | |
| Most people in my community expect men to have the final word about decisions in the home | | | ● | | | | | | ● | |
| Most people in my community do not expect girls and boys to share household tasks equally | | | ○ | | | | | | ● | |
| Most men in my community are the ones who make the decisions in their home | | | ● | | | | | | ● | |
| **Financial and economic empowerment domain** | | | | | | | | | | |
| Most women in my community have the same chance to work outside the home as men | | | | ● | | | | | | ● |
| Most people in my community expect women to have the same chance to work outside the home as men | | | | ● | | | | | | ● |
| **Marriage and relationship domain** | | | | | | | | | | |
| A boy should be able to have a girlfriend if he wants to | | | | ● | | | | | ● | |
| A girl should be able to have a boyfriend if she wants to | | | | ● | | | | | ● | |
| **Sexual and reproductive health domain** | | | | | | | | | | |
| Families should control their daughters' behaviors more than their sons | | ● | | | | | ● | | | |
| Most families in my community control their daughters' behaviors more than their sons' | | ● | | | | | ● | | | |
| Most people in my community expect families to control their daughter's behavior more than their sons' | | ● | | | | | ● | | | |

Grey background = perceived community-level gender norms

White background = gender attitudes

Green highlighted factor loadings = above 0.50 (strong loading)

Yellow highlighted factor loadings = above 0.30 (weak loading)

and low cumulative variance explained. In contrast, the initial five-factor solution showed indications of a five-domains structure (*education; sexual and reproductive health; relationships and marriage; time use; and financial and economic empowerment*). After having refined the five-factor solution, we found that the variables captured their respective domains. However, this was only the case for a reduced set of gender norms items in both datasets. Therefore, we propose an adaptation of the GAGE gender norms scale for Ethiopia and Bangladesh.

Concerning the inadequate two-factor solution, we hypothesise that the distinction between gender attitudes and gender norms in this dataset is not substantial enough to yield two factors in the EFA. This may stem from the community-level gender norms being reported by adolescents themselves, rather than other community members. Consequently, their responses likely reflect their own perspective on community norms rather than the actual norms within the community. Another reason may be that the multidomain structure of the GAGE gender norms scale cannot be captured by a two-factor solution as this would oversimplify the complex structure of the gender norms items.

Evidence for a two-factor solution of gender norms items has been observed in the Nepali context, with the G-Norm scale differentiating between descriptive and injunctive norms [3]. The G-NORM scale has been incorporated into the GAGE gender norms scale, which also contains descriptive and injunctive norms [29]. However, Baird et al.'s analysis does not account for the descriptive-injunctive distinction, instead creating separate scores for individual-level gender attitudes and perceived community-level gender norms.

A recent study among adolescents in Bangladesh also successfully validated a multi-domain structure for a 'gender norms attitude' scale (M-GNAS) [5]. This four-domain structure includes 13 items and covers the areas of gender attitudes identified as dominant among Bangladeshi youth: gender-appropriate behaviour, family financial decisions, family responsibility, and career choice [5]. Islam et al., (2024) [5] highlighted that the lack of women's empowerment in decision-making and deeply entrenched gender norms within the family and socioeconomic context are specific to the Bangladeshi context. This makes the *time-use* variables in the GAGE gender norms scale particularly important.

The exclusion of certain items in the refined five-factor solutions may indicate conceptual issues in the GAGE gender norms scale. One reason could be that these items do not align closely enough with the content represented in the factor structures. The item 'cr_edu_boysfeelings', for example, asks, whether "boys should be able to show their feelings without the fear of being teased". This statement does not only apply to a school context, nor is it similar to the other school-related questions. Yet, the item was assigned to the *education* domain. Furthermore, this item contains two questions in one – whether boys should be able to show their feelings and whether they should be able to do so without fear of being teased. This makes it challenging to interpret the responses. Another conceptual issue is the discrepancy between the content of the items and the domains. In particular, the *sexual and reproductive health* domain includes one item about how girls perceive their bodies in the transition to womanhood, but the remaining items ask about control (e.g., "Families should control their daughters' behaviors more than their sons"), which is not specifically related to sexual and reproductive health. Although sexual and reproductive health questions should be asked with caution when surveying young adolescents, this domain does not cover the content it suggests. To align the questions on control more closely with the domain of sexual and reproductive health, specific behaviours could be asked, such as the choice of contraceptive method or the decision when to have children, etc.

Another limitation of the GAGE gender norms scale could be its focus on a binary understanding of sex, along with gendered questions that are only applicable to one of the two sexes. This conceptual aspect likely applies to most existing gender norms scales. Nevertheless, gender norms could be assessed in a way that makes them applicable to both or all genders. Moreover, gender norms and attitudes in the GAGE gender norms scale are sometimes measured indirectly through other constructs, such as gender roles, gender stereotypes, and traits [5]. For example, items in the *time-use* domain, such as "girls and boys should share household tasks equally", may relate more to the construct of gender roles than gender norms. This reflects the inconsistency in the definition of gender norms, the conflation of terms, and the challenges in the conceptualisation of gender-related aspects in the scientific literature.

Comparing the refined five-factor solutions for Ethiopia and Bangladesh, we found that the 15 gender norms items in Bangladesh are the same 15 gender norms items in Ethiopia, with two additional items specific to the Ethiopia dataset. This demonstrates that a subset of gender norms items was retained consistently across two different geographical contexts. We hypothesise that these variables may be more robust across cultures. Although further research is needed, these variables could serve as a basis for improving the GAGE gender norms scale and for facilitating cross-cultural comparisons. In contrast, the remaining 17 and 15 variables out of the original 30 items indicate that up to half of the items may not be well-suited for the GAGE gender norms scale in Ethiopia and Bangladesh. These items may need to be refined to enhance their alignment with the five-domain structure. This underscores the importance of validating the scale for each cultural context [5,38].

Our results may be generalisable to countries with a similar cultural context to Ethiopia and Bangladesh. Depending on the extent of differences in attitudes to gender norms between different groups of adolescents or regions within a country, it may be useful to conduct EFA separately for subgroups. For example, the Afar and Oromia regions in Ethiopia have been identified for their different and deeply entrenched gender norms [28]. It is possible that similar subgroups of adolescents in different countries could have more similar attitudes to gender norms than different subgroups within a country. While it is recommended to use the list of items that fits best for the specific country or subpopulation (e.g., 15 items for the Bangladesh and 17 items for the Ethiopia dataset), it might be beneficial to use the same set of items when conducting comparisons between countries or subgroups (e.g., 17 items for both datasets) [20].

## 4.2 Strengths and limitations

In our study, we followed methodological recommendations in performing our EFA by using PFA as the conceptually desirable method compared to PCA [33,39], by using oblique rather than orthogonal rotation due to the high correlations among the gender norms items [34], by using relatively large samples (which are particularly important in EFAs) [34] from two LMI countries, and by conducting several factor models [40]. The latter increases the likelihood that the results can be generalised if the same factor structures are retrieved [34]. This may be the case for 15 gender norms items in our analysis.

The limitations of our study are twofold. First, there are limitations in the GAGE datasets which we rely on in our secondary data analysis. Despite being administered by skilled and trained interviewers, the GAGE data may still be influenced by social desirability and other biases, such as measurement invariance. Additionally, the GAGE gender norms items reflect perceived rather than objective measures. Also, some items in the scale may need to be rephrased to produce more valid outcomes.

Furthermore, the provided response options – 'agree', 'partially agree' and 'disagree' - are unbalanced due to the absence of a weak negative response option, such as 'partially disagree'. Recent gender norms scales employ five-point Likert scales, which provide an additional neutral response option [3,5]. Additionally, two factors in the refined five-factor solution (namely *marriage and relationship* and *financial and economic empowerment* norms) each consist of only two variables in both datasets. This is usually considered an insufficient amount for factor analysis. The GAGE data may benefit from additional variables in both domains.

Second, our analysis has its own methodological limitations. We did not try other rotation and extraction methods other than oblimin principal axis factoring in our EFA. Moreover, we did not analyse the factor structures between different subgroups of adolescents considered vulnerable.

To note, factor structures were interpreted based on their factor loadings, communalities, and uniqueness values, as well as their cumulative variance explained. Gender norm scales should not be developed based on statistical findings alone, but should also incorporate theoretical and conceptual considerations. Particularly the 15 and 13 excluded items in the refined five-factor solutions should be culturally and linguistically rephrased to increase cultural validity and avoid misinterpretations. Moreover, items with lower communalities may not align well with the intended domain or categorisation,

or may overlap poorly with other items within the same domain. These items may need to be adapted or assigned to another domain and only items without strong theoretical rationale may need to be excluded.

The rather low cumulative variances explained, in the initial factor solutions, may be attributed to the cultural complexity, context sensitivity, and the multidimensional nature of gender norms. The cultural heterogeneity within Bangladesh and Ethiopia, as well as literacy and interpretation issues among marginalised adolescents in LMI settings may also have contributed. In social sciences, the cumulative variance explained tends to be rather low, particularly when attitudes or perceptions are studied [37].

Lastly, we conducted EFA and not Confirmatory Factor Analysis (CFA). Besides further refining the GAGE gender norms items, conducting CFA or Item Response Theory could be necessary next steps to validate the scale. Further qualitative validation (e.g., cognitive interviewing) could support the process of refining the GAGE gender norms scale and enhance the age- and culture-appropriate adaptation of items.

### 4.3 Conclusion

Gender norms research faces a number of challenges in defining, measuring, and conceptualising gender norms. This applies particularly to gender norms scales for adolescents. Our results indicate that separating the GAGE gender norms items into a two-factor structure with an individual-community distinction may not reflect the data structure for the Ethiopia and Bangladesh datasets. Instead, a multi-domain structure seems more promising. In fact, only our refined five-factor solutions yielded promising results with a reduced set of items. Due to issues with wording or domain assignment of some gender norms items, we propose an adaptation of the GAGE gender norms scale. This could include a cultural and linguistical adaptation, a stronger theoretical foundation and an improved assignment of items to the different domains. With our EFAs, we contribute to improving the validity of a gender norms scale for adolescents in two LMI countries. A refined gender norms scale could be valuable for developing more effective interventions in Ethiopia and Bangladesh to reduce inequitable gender norms.

### Supporting information

**S1 Table. Variable names of the GAGE gender norms items in the Ethiopia and Bangladesh datasets.**
(DOCX)

### Author contributions

**Conceptualization:** Anita Alaze, John Grosser.

**Formal analysis:** Anita Alaze, John Grosser.

**Funding acquisition:** Oliver Razum, Céline Miani.

**Investigation:** Anita Alaze, John Grosser.

**Methodology:** Anita Alaze, John Grosser.

**Supervision:** Céline Miani.

**Visualization:** Anita Alaze, John Grosser.

**Writing – original draft:** Anita Alaze, John Grosser.

**Writing – review & editing:** Oliver Razum, Céline Miani.

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
