## [Decision Letter · Decision Letter 0]

5 May 2025

PGPH-D-25-00488

Assessing the factor structure of the GAGE gender norms scale for adolescents: Exploratory Factor Analyses in datasets from Ethiopia and Bangladesh

Dear Dr. Anita Alaze,

Thank you for submitting your manuscript to PLOS Global Public Health. After careful consideration, we feel that it has merit but does not fully meet PLOS Global Public Health’s publication criteria as it currently stands. Therefore, we invite you to submit a revised version of the manuscript that addresses the points raised during the review process.

We look forward to receiving your revised manuscript.

Kind regards,

Ifunanya Clara Agu

Academic Editor

Journal Requirements:

Additional Editor Comments (if provided):

Reviewers' comments:

Reviewer's Responses to Questions

**Comments to the Author**

1. Does this manuscript meet PLOS Global Public Health’s publication criteria? Is the manuscript technically sound, and do the data support the conclusions? The manuscript must describe methodologically and ethically rigorous research with conclusions that are appropriately drawn based on the data presented.

Reviewer #1: Yes

Reviewer #2: Yes

Reviewer #3: Yes

Reviewer #4: Partly

2. Has the statistical analysis been performed appropriately and rigorously?

Reviewer #1: Yes

Reviewer #2: Yes

Reviewer #3: N/A

Reviewer #4: No

3. Have the authors made all data underlying the findings in their manuscript fully available (please refer to the Data Availability Statement at the start of the manuscript PDF file)?

Reviewer #1: Yes

Reviewer #2: Yes

Reviewer #3: Yes

Reviewer #4: Yes

4. Is the manuscript presented in an intelligible fashion and written in standard English?

Reviewer #1: Yes

Reviewer #2: Yes

Reviewer #3: Yes

Reviewer #4: Yes

5. Review Comments to the Author

Reviewer #1: Title: Assessing the factor structure of the GAGE gender norms scale for adolescents: Exploratory Factor Analyses in datasets from Ethiopia and Bangladesh

General Comments

Overall, this is a well written manuscript evaluating the psychometric structure of the GAGE gender norms scale for adolescents in Ethiopia and Bangladesh by applying Exploratory Factor Analysis to test its proposed two- and five-factor models. The study identifies useful findings for a refined five-factor structure, noting that based on a reduced set of items, it better captures gender norms in Ethiopia and Bangladesh than the original two-factor model. The study suggests the need for adapting and further validating the GAGE gender norms scale for these and similar cultural contexts. Some revisions are proposed and detailed below.

- For a lay audience it would be useful to clarify in the distinction of gender norms (re: stereotypes, roles) & rationale for EFA.

Introduction

- Line 94: If possible, it would be helpful to expand on the effects of gender norms on morbidity and mortality during adolescence. At present, the statement is vague with no direct substantiation.

- Line 97: Similarly, it would be helpful to strengthen the claim for how the adoption of gender inequitable attitudes and roles leads to risks of adverse gender-related health outcomes.

- Line 112: Are all 8 items excluded from this adaptation tied back to the topics of intimate relationships, gender-based violence, and sexual behaviours enumerated above?

- It may be useful to detail more elaborately the concern for gender norm scales developed in the Global North; without due consideration for marginalized youth (lines 119-124), given it defines the research gaps aimed to be filled by the GAGE project.

- Line 125: The GAGE project is the largest longitudinal study focused on adolescents in the Global South, but there is no mention of this scope. Moreover, there is a strong emphasis on adolescents and young key populations (ie. out of school; with disabilities; married under 18, with greater risk for early childbearing; conflict-affected) but no clarification of such detail in the introduction.

- Line 127: If possible, it would be informative to expand on what the GAGE project has drawn from prior scales, and which items have been newly introduced.

- Line 134: If possible, could you clarify what you mean by physical and mental health? The GAGE project is described as holistic in approaching adolescent well-being more broadly, so there requires clarity either on their more limited scope of impact, or elaboration on the facets of physical and mental health referred to here.

- Line 138: The GAGE project has 6 domains (education and learning; health, nutrition, and sexual and reproductive health; bodily integrity and freedom from violence; psychosocial well-being; voice and agency; and economic empowerment) - can you clarify whether the project went on to change these domains, or whether you chose to consolidate them? Can you explain this choice?

Materials and Methods

- Line 155: Can you clarify whether the study was limited to adolescents and their caregivers? To my understanding, the study was meant to consult adults across the adolescents' communities.

- Line 157: The study was intended to focus on ayKP, as opposed to only purposefully including them - not to simply capture diversity but to fill the gap in which they are excluded from the body of research made publicly available. Clarification to this statement would be beneficial.

Discussion

- Line 428: It may be of value to clarify the point around control, and where it could have been more closely aligned to bodily autonomy.

- Line 456: While there is mention of cultural contexts nationally, what are the considerations for how items may need to be refined to adapt to regions within a country?

Reviewer #2: The submitted manuscript presents a relevant and timely contribution to the study of gender norms among adolescents in low- and middle-income countries (LMICs), particularly through the psychometric evaluation of the GAGE gender norms scale. The use of datasets from both Ethiopia and Bangladesh strengthens the cross-cultural relevance of the findings.

Strengths:

The methodological approach, including the use of Exploratory Factor Analysis (EFA) with principal axis factoring and oblique rotation, is appropriate and well executed.

The paper contributes meaningfully to the literature by critically examining the domain structure of a widely used tool (GAGE gender norms scale) and proposing refinements grounded in empirical data.

The authors provide detailed rationale for each step in refining the factor model and appropriately justify decisions to exclude variables.

Concerns and Suggestions:

Conceptual Framing and Clarity:

The manuscript would benefit from a clearer conceptual explanation of why the two-factor model (individual vs. community level) was initially considered and why it failed empirically. While the results are clearly presented, additional theoretical reflection on this mismatch would enhance the discussion.

More elaboration is needed on the implications of the low communalities and uniqueness values. For example, what does this say about the conceptual soundness or cultural validity of certain items?

EFA Interpretation:

While the iterative refinement process is rigorous, it is notable that the final refined five-factor solution for Bangladesh retains only 15 of 30 items, and for Ethiopia, 17 items. This may reduce the scale's comprehensiveness. I recommend more discussion on the trade-off between parsimony and content validity and whether some of the removed items might be culturally rephrased rather than excluded.

The cumulative variance explained (especially in the initial solutions) is quite low. The authors note this, but additional reflection on the adequacy of these thresholds in social science research (and any regional/contextual considerations) would improve the justification.

Recommendations for Future Research:

The conclusion rightly calls for Confirmatory Factor Analysis (CFA), but the authors could also discuss other approaches, such as Item Response Theory (IRT), or qualitative validation (e.g., cognitive interviewing), especially for age- and culture-appropriate adaptation of items.

Language and Structure:

Overall, the manuscript is well written, but a few sentences in the results section are densely packed and could benefit from restructuring for clarity, particularly when presenting loadings and explaining the rationale for item exclusion.

Ethical and Publication Standards:

I found no concerns regarding research ethics, dual publication, or competing interests. The use of secondary data has been appropriately acknowledged and justified.

Recommendation:

I recommend a minor revision. With improved theoretical framing and discussion of methodological implications, this paper will make a valuable contribution to global adolescent health and gender research.

Reviewer #3: Using the Gender and Adolescence: Global Evidence (GAGE) project datasets collected from 2017-2018 in Ethiopia and Bangladesh, the authors adopt Exploratory Factor Analyses approach to assess the factor structure of the GAGE gender norms scale with a focus on vulnerable adolescents.

The authors provide great insights into quantitative measurement of vulnerable adolescent gender norms using a two-factor level of Individual level gender attitude and community level descriptive and injunctive gender norms; as well as five-factor domains of gender norms on basis of education; time use; financial inclusion and economic empowerment; relationships and marriage; and sexual and reproductive health. I have the following comments that could help strengthen the paper:

1. In the second paragraph of the introduction section, the authors provide identified gap on the lack of quantitative measures to assess agreement with gender norms. However, given gender norms are invisible, un/spoken rules on how to behave, this also requires qualitative assessments around them. The paper would benefit from provision of insights on what has been done on qualitative measurements of gender norms, the gaps with such measurements that warranty the need for quantitative measures of gender norms as per the current study.

2. In the materials and methods section, the authors note that GAGE project data captured diversity data of the vulnerable adolescents. Are there any variations across the five-factor level analysis based on the differential nature of adolescents’ vulnerability characterised by disabilities; adolescents who married before age 18; refugees; adolescent mothers and out-of-school youth for Ethiopia and Bangladesh. This could be provided in the results section. In case the data was not analysed across such dimensions, this could be highlighted in the discussion and study limitation sections to help strengthen the paper.

Other minor suggested edits for typos are provided as comments in the attached paper. Looking forward to seeing this amazing research work published in PLOS Global Public Health!

Reviewer #4: General comments of the paper:

The paper on assessing the factor structure of the GAGE gender norms scale read well and is well structured. The work is significant as it explored how gender norms impact adolescents in Ethiopia and Bangladesh. The study aim was clear and was about to validate the GAGE scale, which measures gender norms and their influence on adolescents' lives. The findings inform socio-cultural factors that shape gendered experiences during adolescents especially those with particularities. However, throughout the paper, these considerations were left behind though they were the focus.

Authors can improve their paper by providing the light on the items to be removed and why, show the gaps these items may bring when the GAGE is used as it was developed.

Finally, provide the importance of the final GAGE scale and its generalizability.

Points by points

Pre-section: The Title

The title is too long (The normal length shouldn’t go beyond 15 words and so on) and the GAGE word is not speaking itself in the title (I have seen that it is the abbreviation of Gender and Adolescence: Global Evidence)

You can go by “Gender and Adolescents Global Evidence: Exploratory Factor Analyses of Gender norms in Ethiopia and Bangladesh” Please review if it fits.

Section 1: Abstract

The background section in the abstract is informative. The authors can add also what say literature about the importance to include gender norms in the adolescent’s health trajectories before entering to the Lower- and Middle-income countries to explain how the study relates to the previously published research?

Does the gap remain in the quantitative scales? What qualitative perspective brought in the literature so far? Clarify the difference here or change the language of quantitizing.

The authors also can explain in full all abbreviations and keep the abbreviations in the main text (the body of the paper).

Method: The items composed datasets are not clear for readers. Are these datasets relating to the adolescents as the general population? What age do you consider as adolescent for this study? What was the main test for which outcomes? Clarify and define the research question!!!!!

Results: As the method was not clearly showing the main test, variables and outcome measures, this may be hard for readers to discover the real meaning of the results within the abstract. This should be concise.

Conclusion: This conclusion is clear from the unclear results. The authors may adapt the conclusion with the results presented in the abstract. Also, they can define the suitability in this context as only the word does not clarify what are the five domains in the abstract.

Also, the authors may add the purpose of the study if it was to refine the datasets in order to define the gender norms in both countries

Section 2: Main document

1. Introduction:

Lines 68-74: The paragraph is nice, the authors need to add what literature say about this before entering to what is missing (Gaps identified).

Lines 75-: Do speak only on the quantitative measures, what about qualitative? The authors can suggest literature review about this.

Lines 85-86 highlight the existing gender norms scales, could you show the gaps about how they were conceptualized and the impacts they have on gender norms for readers???

Lines 142-148: The authors highlighted the aim as to explore the not yet validated GAGE gender norms statistically which are published already. Clarify the readers that though the previous gender norms are published, but they were not probably tested for their psychometric measurement or show the low internal consistency which motivated you to explore further statistically.

Lines 147-148: Share the results from this computation

No research question defined apart from the purpose

2. Materials and Methods

Line 150: Ethical consideration

Talk about the data handling privacy though the study did not require ethical clearance.

Line 153: Data

Lines 154-157: Explain that you did not correct any data; you used the existing program’s data to shape the GAGE (Declare any relationship with you and the project and possible conflict of interest according to the journal guidelines).

Lines 157-160: Bring in a new concept which is vulnerability (disability, early marriage, refugees, adolescent mothers and those dropped-out from school) and was not introduced from the beginning even in the abstract. Be consistent throughout and this consideration should be explained and tied from the beginning.

Line 171: Measures

In this part, there are two outcome measures. The authors may opt to define first these two and their compositions to clarify the readers what are they.

Lines 174-175: Where comes education domain??? Name all domains and explain all including the items under each domain (you can simply them in a table).

Lines 187-190: Clarify the questions in the brackets

Line 191: Exploratory analysis

Lines 192: Explain the choice of using exploratory analysis

Line 196-197: Explain the exact test for suitability and the level of acceptability for the suitability (Provide interval measures)

Lines 205-207: What are the results from computation before refining the five-factor solution? Explain for readers

Lines 209-212: For the cut-off, how many variables eliminated as fallen under or equal to cut-off measure, how many times did you repeated the process to arrive at the acceptable loading?

Line 213: Results

Lines 215-219: Explain the difference in the sample for both countries. Why don’t you consider equal sample size for the two countries (equal male and equal female)?

Explain if the table was generated by the authors or it is from the GAGE project

Lines 222-232: Some information is repeated from above; we computed Bartlett’s test…we refined the factors… (line 231). The author can reduce the duplicates in the sections.

Line 239: Two factor solution: Is this the final table after all loadings? The value of less than 0.30 is prevalent; what is your take? Especially in grey color. Will this remain? Remember the cut-off.

Revise all tables and define the cut-off from within.

I am also afraid of the influence of the high number (N) within the population which may influence the decision.

Line 323: Refined five-factor solution

For the refined five-factor solution, explain whether the removed items in Ethiopia were also removed from Bangladesh dataset.

Any common cumulative variance for both countries???

Explain table 8 with similarities and differences for both countries

Line 387: Discussion

Good discussion flow. The authors can separate all five domains in their discussion for a deep reading with clarity.

Line 405: As evidence were observed in the other countries rather than Ethiopia and Bangladesh, the discussion here can focus on the important findings from this study

Line 457: Strengths and Limitations

Line 456: Speak on the limitation about the triangulation and generalizability of the present findings in the GAGE countries rather than Ethiopia and Bangladesh as some items were removed

Line 485: Conclusion

Revise the entire conclusion to show the importance of the work done so far.

6. PLOS authors have the option to publish the peer review history of their article (what does this mean?). If published, this will include your full peer review and any attached files.

**Do you want your identity to be public for this peer review?** For information about this choice, including consent withdrawal, please see our Privacy Policy.

Reviewer #1: No

Reviewer #2: **Yes: **Arz O Sama

Reviewer #3: **Yes: **Millicent L. Liani

Reviewer #4: No

While revising your submission, please upload your figure files to the Preflight Analysis and Conversion Engine (PACE) digital diagnostic tool, https://pacev2.apexcovantage.com/. PACE helps ensure that figures meet PLOS requirements. To use PACE, you must first register as a user. Registration is free. Then, login and navigate to the UPLOAD tab, where you will find detailed instructions on how to use the tool. If you encounter any issues or have any questions when using PACE, please email PLOS at figures@plos.org. Please note that Supporting Information files do not need this step

---

## [Decision Letter · Decision Letter 1]

26 Aug 2025

How does a recent gender norms scale perform? Exploratory Factor Analyses among adolescents in Ethiopia and Bangladesh

PGPH-D-25-00488R1

Dear Anita Alaze,

We are pleased to inform you that your manuscript 'How does a recent gender norms scale perform? Exploratory Factor Analyses among adolescents in Ethiopia and Bangladesh' has been provisionally accepted for publication in PLOS Global Public Health.

Best regards,

Ifunanya Clara Agu

Academic Editor

Reviewer Comments (if any, and for reference):

Reviewer's Responses to Questions

**Comments to the Author**

1. If the authors have adequately addressed your comments raised in a previous round of review and you feel that this manuscript is now acceptable for publication, you may indicate that here to bypass the “Comments to the Author” section, enter your conflict of interest statement in the “Confidential to Editor” section, and submit your "Accept" recommendation.

Reviewer #2: All comments have been addressed

Reviewer #3: All comments have been addressed

Reviewer #4: All comments have been addressed

2. Does this manuscript meet PLOS Global Public Health’s publication criteria? Is the manuscript technically sound, and do the data support the conclusions? The manuscript must describe methodologically and ethically rigorous research with conclusions that are appropriately drawn based on the data presented.

Reviewer #2: Yes

Reviewer #3: Yes

Reviewer #4: Yes

3. Has the statistical analysis been performed appropriately and rigorously?

Reviewer #2: Yes

Reviewer #3: Yes

Reviewer #4: Yes

4. Have the authors made all data underlying the findings in their manuscript fully available (please refer to the Data Availability Statement at the start of the manuscript PDF file)?

Reviewer #2: Yes

Reviewer #3: Yes

Reviewer #4: Yes

5. Is the manuscript presented in an intelligible fashion and written in standard English?

Reviewer #2: Yes

Reviewer #3: Yes

Reviewer #4: Yes

6. Review Comments to the Author

Reviewer #2: (No Response)

Reviewer #3: Thank you for the opportunity to review this manuscript. I hereby acknowledge that the authors have addressed all the comments as per the first round of the review. Delighted to see this paper published in PLOS Global Public Health!

Reviewer #4: Well done with the paper. It is now clear on my side

Great job

7. PLOS authors have the option to publish the peer review history of their article (what does this mean?). If published, this will include your full peer review and any attached files.

**Do you want your identity to be public for this peer review?** For information about this choice, including consent withdrawal, please see our Privacy Policy.

Reviewer #2: **Yes: **Arz-O-Sama

Reviewer #3: **Yes: **Millicent L. Liani

Reviewer #4: **Yes: **Dr. Marie Grace Sandra Musabwasoni
